# Risk-Reducing Mastectomy and Reconstruction Following Prophylactic Breast Irradiation: Hope Sustained

**DOI:** 10.3390/cancers13112694

**Published:** 2021-05-30

**Authors:** Merav A. Ben David, Ella Evron, Adi F. Rasco, Ayelet Shai, Benjamin W. Corn

**Affiliations:** 1Assuta Medical Center, Tel Aviv 6971028, Israel; meravak@assuta.co.il; 2Faculty of Health Sciences, Ben-Gurion University of the Negev, Beer Sheva 84105, Israel; 3Oncology, Kaplan Medical Institution, Rehovot 7661041, Israel; 4Faculty of Medicine, The Hebrew University, Jerusalem 91120, Israel; Ben.w.corn@gmail.com or; 5Shamir Medical Center and Sackler School of Medicine, Tel-Aviv University, Tel Aviv 7033001, Israel; rascoa@shamir.gov.il; 6Department of Oncology, Galiliee Medical Center, Naharia 22100, Israel; AyeletS@gmc.gov.il; 7Azrieli Faculty of Medicine, Bar Ilan University, Ramat Gan 5290002, Israel; 8Shaare Zedek Medical Center, Jerusalem 9103102, Israel

**Keywords:** breast cancer, BRCA mutation, prophylactic breast irradiation

## Abstract

**Simple Summary:**

In this study we report the outcome of salvage mastectomy and reconstruction in 11 BRCA mutation carrier patients that participated in a clinical trial of prophylactic contralateral breast irradiation and suffered reoccurrences of breast cancer in either the ipsilateral or contralateral breast or elected to have the procedure for risk reduction. Patients’ satisfaction and physicians’ assessment of the cosmetic outcome were not inferior for previously irradiated compared to non-irradiated breasts. These results are encouraging and support continuing research as well as a discussion of risk-reduction alternatives besides mastectomy, including prophylactic breast irradiation, in BRCA1/2 mutation carriers.

**Abstract:**

Risk-reducing mastectomy (RRM) is often advocated for BRCA1/2 mutation carriers who face a heightened lifetime risk of breast cancer. However, many carrier patients seek alternative risk-reducing measures. In a phase II nonrandomized trial, we previously reported that prophylactic irradiation to the contralateral breast among BRCA carriers undergoing breast-conserving treatment significantly reduced subsequent contralateral breast cancer. Herein, we report the outcome of salvage mastectomy and reconstruction in 11 patients that suffered reoccurrences of breast cancer in either the ipsilateral or contralateral breast or elected to have the procedure for risk reduction during the eight-year follow-up period. Patients’ satisfaction with the procedure and physicians’ assessment of the cosmetic outcome were not inferior for previously irradiated compared to non-irradiated breasts. Although the numbers are small, the results are encouraging and sustain hope in a challenging population. Our findings support continuing research as well as a discussion of risk-reduction alternatives besides mastectomy, including prophylactic breast irradiation, in BRCA1/2 mutation carriers.

## 1. Introduction

BRCA mutations are particularly prevalent in Israel, as 2.5% of the Ashkenazi Jewish population, which comprises about 50% of Jewish Israelis, carry a specific founder mutation in BRCA1 or BRCA2. Accordingly, 20% of young-age (<40) breast cancers that arise in Israeli women are attributed to these mutations [1,2]. The management of BRCA-associated breast cancer, therefore, presents recurring challenges for Israeli oncologists. Women with BRCA-associated breast cancer often present at a relatively young age with decidedly more aggressive cancers [3]. Moreover, they face a high risk of developing contralateral breast cancer as well as ovarian cancer [4]. Consequently, mastectomy of the diseased breast as well as risk-reducing mastectomy (RRM) of the contralateral breast are, by necessity, part of the treatment discussion with these patients. Notwithstanding, the decision to remove one’s breasts is not easy, and behooves intimate discourse between a woman, her body, and her “womanhood”. Therefore, additional risk-reducing measures are needed for BRCA mutation carriers who forgo mastectomy.

Between 2007 and 2017, 162 BRCA mutation carriers with unilateral breast cancer were enrolled in a national phase II nonrandomized trial, where the patients had the option of prophylactic contralateral breast irradiation (Appendix A). At a median follow up of five years, 10 patients developed contralateral breast cancer in the control arm as compared with two patients in the intervention arm (log-rank *P* = 0.011) [5]. The addition of contralateral breast irradiation was associated with a significant reduction of subsequent contralateral breast cancers and a delay in their onset. Longer follow up is needed and continues [6]; nevertheless, some investigators have posited that the data already justify offering bilateral irradiation to BRCA carriers undergoing treatment for breast cancer [7]. To the best of our knowledge, no other studies are investigating contralateral prophylactic radiotherapy in hereditary breast cancer patients (clinicaltrials.gov, accessed 14 May 2020). During the follow-up period, a minority of the patients experienced subsequent ipsilateral or contralateral breast cancer necessitating salvage procedures, and few patients elected to pursue risk-reducing mastectomy plus reconstruction. 

It has been suggested that prior breast irradiation may increase complications and jeopardize cosmesis of subsequent breast reconstruction [8]. The optimal timing and technique of breast reconstruction in patients who need chest-wall irradiation (PMRT) is controversial [8,9], and it has been repeatedly reported that chest wall irradiation is associated with inferior reconstructive results. Thus, in patients undergoing implant-based reconstruction, PMRT increases rates of infections, capsular contracture, implant loss and overall reconstructive failure requiring revision surgeries, whereas in patients undergoing autologous reconstruction, PMRT has been associated with fibrosis, distortion of breast shape, volume loss, and fat necrosis. Moreover, patient-reported outcomes after reconstruction are lower in women receiving PMRT [8]. Therefore, we elected to carefully review and report the outcome of breast reconstruction among trial participants that underwent bilateral mastectomy and reconstruction during the follow-up period.

## 2. Methods

Permission to proceed with the aforementioned analysis was obtained from the Institutional Review Board (IRB) at Kaplan Medical Center (Rehovot, Israel). We systematically contacted the trial participants that had mastectomy and reconstruction during the follow-up period and requested their consent for participation in this secondary analysis. Patients who were willing to participate were invited to return to the clinic for an evaluation visit that included photo-documentation. We asked patients who were not willing to come, especially in view of the COVID-19 pandemic, to consent to an interview by phone and to submit their own photos. We used the Baker score—a validated four-point rating scale ranging from excellent to poor cosmetic result—for evaluation of the reconstruction on both sides [10]. We instructed the patients to score their satisfaction from 1 to 10, where 10 is maximal satisfaction. The treating physicians scored their impression using the Baker score, based on their physical examination (when performed) or based on the pictures that were submitted by the patients. In most cases, multiple photos, including anterior and lateral pictures, were provided. We then selected the most representative photographs for display in Table 1 and Table 2, respectively. The choice of plastic surgeon and type of reconstruction surgery were determined by the patient and her physician, respectively. All the photos and satisfaction scores were obtained recently in preparation for this report. 

## 3. Results

Five patients in the experimental arm who received contralateral risk-reducing irradiation underwent subsequent bilateral mastectomy and reconstruction, three following contralateral breast cancer (pt. 1B, 2B, 4B), one after ipsilateral cancer recurrence (pt. 3B), and one for risk reduction without contralateral disease, as seen in Table 1. The latter patient declined to participate in this report or to submit pictures. The median time between bilateral irradiation and surgery was 5.5 years (range five to seven years). 

Twelve patients in the control group who received standard treatment to the ipsilateral breast only underwent subsequent mastectomy and reconstruction of the contralateral breast, three following contralateral breast cancer (pt. 4C, 6C, and one patient that did not wish to participate or send pictures), one patient after ipsilateral cancer recurrence (pt.2C), and eight patients for risk reduction without disease (pt.1C, 3C, 5C, 7C, and four patients that did not wish to participate or send pictures), as seen in Table 2. The median time between ipsilateral breast irradiation and surgery was two years (range one to five years).

The average Baker score was 2.14 both for reconstruction after breast irradiation (14 breasts) and for reconstruction without previous breast irradiation (seven breasts). Most patients in both groups were pleased with their reconstruction outcome (average 7.64/10 for reconstruction after prior prophylactic irradiation and 7.29/10 for reconstruction without prior breast irradiation)—see Table 3—implying that prior breast irradiation did not compromise subsequent reconstruction results. One patient in the experimental arm (prior prophylactic irradiation) and five in the control arm who had mastectomy and reconstruction during the follow-up period refused to participate in this report or to submit pictures. 

## 4. Discussion

Women who carry a BRCA1/2 germ line mutation face a high lifetime risk for the development of breast and ovarian cancer [11,12,13]. Thus, the overall lifetime risk of breast cancer for women carrying a BRCA1/2 mutation is between 50% and 80%, while the lifetime risk for ovarian cancer is 15% to 40% [14]. Patients harboring BRCA1/2-mutated breast cancer tend to have an earlier age at onset—usually before 50 years of age, particularly for BRCA1 cancers, and a higher risk for contralateral breast cancer [4,15]. Consequently, bilateral salpingo-oophorectomy (BSO) and risk-reducing mastectomy (RRM) are often advocated for these women. While RRM markedly decreases the occurrence of breast cancer in BRCA mutation carriers, its effect on survival is less apparent [16], especially in patients who already developed breast cancer and face the hazard of systemic recurrence [17]. Notably, many BRCA carriers pursue alternative preventive measures because of fears related to detrimental effects of RRM on sensation, body image and sexuality [18]. Therefore, additional risk-reducing interventions are needed for carriers interested in breast conservation. For BRCA carriers who already developed breast cancer, most studies found that the outcome of breast-conserving therapy (lumpectomy and whole breast radiotherapy) is comparable to non-carriers [19,20,21], but their risk of subsequent cancer in the contralateral breast is markedly increased [22,23]. Thus, it was recently reported that the 10-year cumulative risk of a second contralateral breast cancer was 23.9% in carriers diagnosed before age 41 and 12.6% in carriers who were first diagnosed at 41 to 49 years [22]. Given the propensity of Ashkenazi Jewish women to harbor BRCA mutations [2], the state of Israel has an enriched population with a unique risk profile. Accordingly, a national effort was launched in that country which engaged the majority of the Israeli oncologic community to plan and implement a trial to deal with the clinical, emotional and ethical ramifications of this disease. In our phase II nonrandomized trial, we found that prophylactic irradiation to the contralateral (ostensibly healthy) breast in BRCA carriers undergoing breast conserving treatment significantly reduced subsequent contralateral breast cancer [5]. Given the finite risk of failure, concern has been raised about the outcome of salvage and reconstructive procedures in the event that subsequent ipsilateral or contralateral cancer develop, as it has been suggested that antecedent breast irradiation may increase complications and jeopardize cosmesis of subsequent breast reconstruction [8]. Herein, we report 11 participants on the trial that had bilateral mastectomy and reconstruction within the follow-up period, including 14 breast reconstructions following breast irradiation and seven reconstructions of non-irradiated breasts. These results suggest that in this patient population of BRCA mutation carriers, previous breast irradiation does not compromise subsequent mastectomy and breast reconstruction. Admittedly, the numbers are small, and the assessment of reconstruction outcome was subjective and reflects the patients’ own experience and judgment of the treating physicians, mostly based on photos. However, the results are encouraging and may reassure high-risk patients who wish to preserve their breasts and choose prophylactic breast irradiation as a risk-reducing modality. Finally, the inclusion of actual photos gives BRCA carriers and the medical community the opportunity to directly examine the results of salvage RRM plus reconstruction and should promote the discussion of alternative primary risk-reducing options.

Genomic insights have provided opportunities to tailor therapies to specific populations. Although the majority of prospective trials in the era of precision medicine have been oriented around systemic therapies [24], radiation strategies should also be incorporated into such meticulous approaches to oncologic management. Indeed, while the genomic revolution has affected the delivery of immunotherapy, biological agents and chemotherapies, it has yet to be integrated into radiation-related strategies. A fundamental tenet of precision medicine is that therapies for cancer should be calibrated with tumor biology [24,25]. Although several recurrence risk signatures are summoned to guide the management of patients with node-negative breast cancer, only recently has preliminary validation of radiosensitivity molecular signatures emerged in the setting of breast cancer [26]. The use of prophylactic irradiation is admittedly controversial but seems to offer an effective alternative to healthcare providers as well as patients struggling with the clinical reality of BRCA mutations. Scrupulous reporting of outcomes must continue to be a manifestation of the moral commitment that investigators make to both of these stakeholders as radiation strategies become informed by biological principles.

Finally, we wish to emphasize that the integration of prophylactic irradiation in the management of the uninvolved breast in the setting of BRCA mutation carriers diagnosed with early-stage breast cancer constitutes a source of hope for women beset by the fears associated with this condition. Prophylactic irradiation of the breast is unencumbered by the significant implications for sexuality and body image that are associated with risk-reducing mastectomy. In this context, we do not refer to hope as an innate characteristic, but rather as a cognitive construct that can assist patients through the trajectory of illness [27]. A recent report has demonstrated that patients with breast cancer, as well as the oncologists who care for them, can be taught to enhance such hopefulness [28]. The advent of creative, non-invasive options in the management of a genomically defined subset of breast cancer constitutes a welcome source of optimism for the concentric circles of patients and caregivers, as well as medical professionals.

## 5. Conclusions 

Our findings are encouraging and support continuing research and discussion of risk-reduction alternatives, including prophylactic breast irradiation, in BRCA1/2 mutation carriers.

## Figures and Tables

**Table 1 cancers-13-02694-t001:** Mastectomy and reconstruction after prophylactic breast irradiation.

#	Year of Birth	Irradiation Therapy (AGE)	Reconstruction Reason, Type	Satisfaction Patients 1-10	Baker Score	Photo
1B	1972	Bilateral 2013 (41)	2019 CLT DCIS SSM Silicon	Rt-10/10 Lt. 7/10	Rt-1 Lt-1	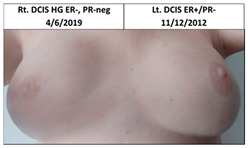
2B	1964	Bilateral 2015 (51)	2020 CLT Ca. NSSM Silicon	Rt-10/10 Lt-10/10	Rt-1 Lt-1	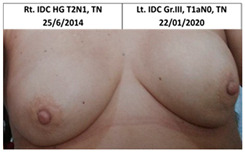
3B	1947	Bilateral 2010 (63)	2015 Local Rec. SSM Silicon	Rt-7/10 Lt-7/10	Rt-2 Lt-3	Lt: IDC Gr.III, T1N0,TN 29/7/2009 Lt Rec: IDC Gr.III, T1N0,TN 6/11/2014
						Pt. declined to have her picture published
4B *	1972	Bilateral 2010 (38)	2017 CLT Ca. SSM Silicon	Rt-4/10 Lt-3/10	Rt-4 Lt-4	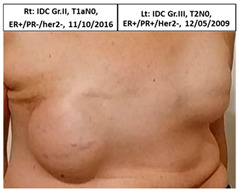

* This patient was a heavy smoker and has severe scoliosis. CLT—contralateral; DCIS—ductal carcinoma in situ; Ca—cancer; Local Rec.—local recurrence; SSM—skin-sparing mastectomy; NSSM—nipple- and skin-sparing mastectomy.

**Table 2 cancers-13-02694-t002:** Mastectomy and reconstruction of non-irradiated breasts.

#	Year of Birth	Irradiation Therapy(AGE)	ReconstructionReason, Type	SatisfactionPatients1-10	Baker Score	Photos
1C	1966	Lt. breast2010(44)	2012Risk Reduc.SSM Silicon	Rt-8/10Lt-8/10	Rt-1Lt-1	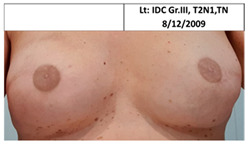
2C	1965	Lt. breast2010(45)	2015Local Rec.SSM Silicon	Rt-7/10Lt-9/10	Rt-4Lt-2	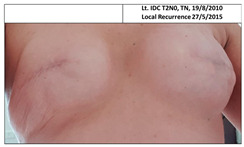
3C	1965	Rt. Breast2011(46)	2012Risk Reduc.SSM Silicon	Rt-8/10Lt-8/10	Rt-2Lt-1	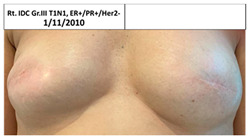
4C	1970	Lt. breast2011(41)	Lt- 2011Rt- 2013NSSM SiliconBil. 2017- tear of implant	Rt-3/10Lt-6/10	Rt-3Lt-2	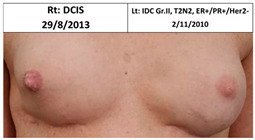
5C	1974	Lt. breast2014(40)	2015Risk Reduc.SSM Silicon	Rt-10/10Lt-10/10	Rt-1Lt-2	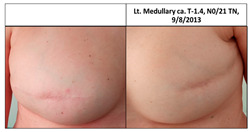
6C	1973	Rt. Breast2016(43)	2018CLT Ca.SSM Silicon	Rt-5/10Lt-6/10	Rt-4Lt-3	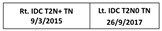
Pt. declined to have her picture published
7C	1978	Lt. Breast2017(39)	2019Risk Reduc.NSSM Silicon	Rt-9/10Lt-9/10	Rt-2Lt-2	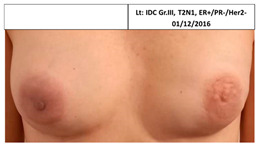

Risk Reduc.—risk reduction; CLT—contralateral; DCIS—ductal carcinoma in situ; Ca—cancer; Local Rec.—local recurrence; SSM—skin-sparing mastectomy; NSSM—nipple- and skin-sparing mastectomy.

**Table 3 cancers-13-02694-t003:** Patients’ satisfaction and Baker score for breast reconstruction after prior breast irradiation (left) versus not (right).

Reconstruction After Irradiation	Breast Cancer	Time from Irradiation to Reconstruction	Patient Satisf. 1–10	Baker Score	Reconstruction No Irradiation	Breast Cancer	Patient Satisf. 1–10	Baker Score
1B Lt	2012	6y 5mo	7	1				
2B RT	2014	5y	10	1				
2B Lt	2020	5y	10	1				
3B Rt	No	4y 9mo	7	2				
3B Lt	2009, 2014	4y 9mo	7	3				
4B Rt	2016	7y 5mo	4	4				
4B Lt	2009	7y 5mo	3	4				
1C Lt	2009	1y 7mo	8	1	1C Rt	No	8	1
2C Lt	2010, 2015	5y 1mo	9	2	2C Rt	No	7	4
3C Rt	2010	7mo	8	2	3C Lt	No	8	1
					4C Rt	2013	3	3
5C Lt	2013	12mo	10	2	5C Rt	No	10	1
6C Rt	2015	2y 3mo	5	4	6C Lt	2017	6	3
7C Lt	2016	1y 7mo	9	2	7C Rt	No	9	2
Average			7.64	2.14	Average		7.29	2.14

B—bilateral irradiation; C—control-standard locoregional treatment.

## Data Availability

The data presented in this study are available in this article (and Appendix A).

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
