# Peer review of "Risk-Reducing Mastectomy and Reconstruction Following Prophylactic Breast Irradiation: Hope Sustained"

_cancers, 2021, doi:10.3390/cancers13112694_

Round 1
Reviewer 1 Report
This brief report compares the cosmetic outcomes and treatment satisfaction in patients who underwent breast reconstruction after prophylactic irradiation vs those patients which not received an irradiation preceding mastectomy.
This treatment approach appears interesting and the presented results might also suggest that prophylactic irradiation is possible without compromising cosmetic outcomes.
It must be taken positively into account that - due to Covid19 - only small patient numbers could be recruited and we fully approve the use of photos send in by the patients themselves.
However, it is necessary to revise several parts of the work. Please check the manuscript for spelling, punctuation and grammatical errors.
- Introduction: The sentences regarding women "without long-term relationships" might be considered insensitive
- Methods: Describe the used baker score - consider carefully, whether the baker score represents an objective assessment tool, it rather seems to be an entirely subjective visual assessment (in contrast to e.g. the BCCT.core software)
- Results:
- patient-numbers and groups are difficult to grasp. This leads to difficulties in understanding the exact amount of patients and therefore the validity of the study. For example, patients in one group could have a capital A, B, C... to allow for easy understanding. Furthermore, a small table showing patient numbers per group who completed the data obtainment would be helpful.
- The tables appear somewhat preliminary - please rearrange for the sake of clarity
- "Satisfaction" with the therapy appears quite general - it would have been more appropriate to ask about different items like appearance, elasticity, consistency, pigmentation etc.. - please consider to rephrase (e.g. 'subj. treatement success (physician).
- Please briefly compare physician and patient assessed cosmetic outcomes. You do not mention the physician-assessed results.
- "However, inclusion of these patients in this analysis is unlikely to have biased the results of the experimental group."
- This stands in conflict with the sentence mentioned directly before and a somewhat clumsy or unsophisticated wording. Perhaps, try "... might have altered the final results"
- Discussion:
- As mentioned above, the Baker-Score is not objective but subjective. We advise to describe this in the discussion and to highlight opportunities for objective assessments in future studies by citing, for example, the following articles:
- https://doi.org/10.3390/cancers12092444
- https://doi.org/10.1016/j.cmpb.2015.11.010
- As mentioned above, the Baker-Score is not objective but subjective. We advise to describe this in the discussion and to highlight opportunities for objective assessments in future studies by citing, for example, the following articles:
Author Response
This brief report compares the cosmetic outcomes and treatment satisfaction in patients who underwent breast reconstruction after prophylactic irradiation vs those patients which not received an irradiation preceding mastectomy.
This treatment approach appears interesting and the presented results might also suggest that prophylactic irradiation is possible without compromising cosmetic outcomes.
It must be taken positively into account that - due to Covid19 - only small patient numbers could be recruited and we fully approve the use of photos send in by the patients themselves.
This report is a sub-analysis of a clinical trial in which 80 BRCA carrier patients received prophylactic irradiation to the contralateral breast. Only 5 of these 80 underwent subsequent bilateral mastectomy and reconstruction, and 13 of 81 in the control group.
Admittedly, these are small numbers, but to the best of our knowledge there are no other trials that tested prophylactic breast irradiation.
However, it is necessary to revise several parts of the work. Please check the manuscript for spelling, punctuation and grammatical errors.
- Introduction: The sentences regarding women "without long-term relationships" might be considered insensitive
- We have removed the sentence in question and apologize for any insensitivity that may have been conveyed.
- Methods: Describe the used baker score - consider carefully, whether the baker score represents an objective assessment tool, it rather seems to be an entirely subjective visual assessment (in contrast to e.g. the BCCT.core software)
- Thank you. Indeed, the Baker score is subjective- based on the appearance of the reconstructed breast. Most patients in this trial sent their own photos and had a telephone interview. The physicians scored the cosmesis based on the physical exam when performed or based on the photos. We changed in “methods” accordingly and took out “physician satisfaction” both from the text and from the tables. In “methods”: we took out "objective".
- Results:
- patient-numbers and groups are difficult to grasp. This leads to difficulties in understanding the exact amount of patients and therefore the validity of the study. For example, patients in one group could have a capital A, B, C... to allow for easy understanding. Furthermore, a small table showing patient numbers per group who completed the data obtainment would be helpful.
- The tables appear somewhat preliminary - please rearrange for the sake of clarity
We revised the tables, altered the numbering and thus attempted to clarify the data.
- "Satisfaction" with the therapy appears quite general - it would have been more appropriate to ask about different items like appearance, elasticity, consistency, pigmentation etc.. - please consider to rephrase (e.g. 'subj. treatement success (physician).
- Most patients in this trial sent their own photos and had a telephone interview. The physicians scored the appearance based on the photos. Unfortunately, we did not assess elasticity, consistency or other physical parameters. "Patient's satisfaction" is a subjective measure that might address all these parameters including cosmesis in her eyes.
- Please briefly compare physician and patient assessed cosmetic outcomes. You do not mention the physician-assessed results.
- We changed in “methods” accordingly and took out “physician satisfaction” both from the text and from the tables.
- "However, inclusion of these patients in this analysis is unlikely to have biased the results of the experimental group."
- This stands in conflict with the sentence mentioned directly before and a somewhat clumsy or unsophisticated wording. Perhaps, try "... might have altered the final results".
- We have removed this comment.
- Discussion:
- As mentioned above, the Baker-Score is not objective but subjective. We advise to describe this in the discussion and to highlight opportunities for objective assessments in future studies by citing, for example, the following articles:
- We revised the discussion and added:
Admittedly the numbers are small, and the assessment of reconstruction outcome was subjective and reflects the patients' own experience and judgment of the treating physicians, mostly based on photos. Yet, the results are encouraging and may reassure high-risk patients who wish to preserve their breasts and choose prophylactic breast irradiation as a risk-reducing modality.

Reviewer 2 Report
The authors have addressed my questions and comments. The revisions have significantly improved the manuscript prior to publication.
Author Response
The authors have addressed my questions and comments. The revisions have significantly improved the manuscript prior to publication.
(No comments by this reviewer)
Thank You

Reviewer 3 Report
Dear authors, I have reviewed your manuscript entiteld: “Risk-Reducing Mastectomy and Reconstruction following Prophylactic Breast Irradiation: Hope Sustained“ and have the following comments:
Abstract:
Please provide some numbers in the abstract. i.e. cohort size, FU time, p-values etc. so that your results cn be put into context.
Introduction:
- Please add a number to particularly prevalent – how many women in % are BRCA carriers in Israel – also define young age breast cancer in brackets (>40a?, >50a?)
Methods:
- Please provide information on the surgeons – were the women operated on by the same surgery team, same site? What was the experience of the surgeons?
Results:
- Please include a patient flow chart
- Table 1 – Please add reasons for reconstruction in a separate column; add range to the results and a 95% confidence interval
- Were irradiation protocols identical in terms of dosage received?
- Please provide a measure of tumor extension in the women that had to undergo contralateral mastectomy due to BC
- Please add patient age at time of irradiation and reconstruction
- Plase add patient BMI
- Please add surgical procedure that was chosen – it is evident from the images that the procedures were different (i.e. some underwent nipple sapring mastectomies, while others didn’t)
- Please provide a median as well as a 95% CI interval also for time interval between irrradiation and surgery.
Discussion: Please discuss the cohort size more critically. It is a really small cohort.
General: While overall interesting, I find this cohort too small to draw any conclusions or to encourage patients to opt for this choice. I would therefore highly recommend to provide more data in a statistically meaningful and comparable way and stick to an objective description of this cohort instead of trying to draw any conclusions. Based on the number of cases your results may be true, random or a result of bias, but I would not recommend to base any clinical recommendation on this very limited dataset.
Kind regards,
Reviewer 3
Author Response
Dear authors, I have reviewed your manuscript entiteld: “Risk-Reducing Mastectomy and Reconstruction following Prophylactic Breast Irradiation: Hope Sustained“ and have the following comments:
Abstract:
Please provide some numbers in the abstract. i.e. cohort size, FU time, p-values etc. so that your results cn be put into context.
We revised the abstract to include numerical content:
Herein, we report the outcome of salvage mastectomy and reconstruction in 11 patients that suffered reoccurrences of breast cancer in either the ipsilateral or contralateral breast or elected to have the procedure for risk reduction during the 8 years follow-up period. Patients’ satisfaction with the procedure and physicians’ assessment of the cosmetic outcome were not inferior for previously irradiated compared to non-irradiated breasts.
Introduction:
- Please add a number to particularly prevalent – how many women in % are BRCA carriers in Israel – also define young age breast cancer in brackets (>40a?, >50a?)
Revised: BRCA mutations are particularly prevalent in Israel, as 2.5% of the Ashkenazi Jewish population, which comprises about 50% of Jewish Israelis, carry a specific founder mutation in BRCA1 or BRCA2. Accordingly, 20% of young age (<40) breast cancers that arise in Israeli women are attributed to these mutations.
Methods:
- Please provide information on the surgeons – were the women operated on by the same surgery team, same site? What was the experience of the surgeons?
Added:
The choice of plastic surgeon and type of reconstruction surgery were determined by the patient and her physician, respectively.
Results:
- Please include a patient flow chart
- Added: supplemental 1
- Table 1 – Please add reasons for reconstruction in a separate column; Added
- add range to the results and a 95% confidence interval.
- Because the numbers are so small, we think that strict statistics may not be meaningful, therefore we modified the report to be more descriptive and “semiquantitative” …
- Were irradiation protocols identical in terms of dosage received?
- Yes, we added Supplemental Fig.1.
Thorough description of the radiation treatment is included in the original paper in: Evron E. et al. Prophylactic irradiation to the contralateral breast for BRCA mutation carriers with early-stage breast cancer. Ann Oncol 2019; 30: 412-417, Ref 5.
- Please provide a measure of tumor extension in the women that had to undergo contralateral mastectomy due to BC.
- This appears above the photos of the relevant breasts.
- Please add patient age at time of irradiation and reconstruction Done
- Please add patient BMI; Unfortunately, we do not have this measure.
- Please add surgical procedure that was chosen – it is evident from the images that the procedures were different (i.e. some underwent nipple sapring mastectomies, while others didn’t); Done
- Please provide a median as well as a 95% CI interval also for time interval between irrradiation and surgery. Added to Results.
Discussion: Please discuss the cohort size more critically. It is a really small cohort.
General: While overall interesting, I find this cohort too small to draw any conclusions or to encourage patients to opt for this choice. I would therefore highly recommend to provide more data in a statistically meaningful and comparable way and stick to an objective description of this cohort instead of trying to draw any conclusions. Based on the number of cases your results may be true, random or a result of bias, but I would not recommend to base any clinical recommendation on this very limited dataset.
We revised the discussion accordingly.

Round 2
Reviewer 1 Report
Dear authors,
Although you have not entirely adressed all of the suggested improvements, your short report has gained in quality and can therefore be considered for publication.
Reviewer 3 Report
Dear authors,
I have rereviewed your manuscript. I am still concerned by the very small cohort size, which you draw your conclusions from, but appreciate your effort that you tried to substantiate some of the statements of your paper with numbers. As you indicate yourself even indicating 95%-CI is not meaningful for this size of a cohort. I find this problematic and would recommend to really rephrase the discussion more strongly in this context. It is a noble goal to publicize images of cosmetic results after this procedure to give women and their treating physicians something to refer to when considering this option. However, I would strictly leave it at that and not draw any conclusions. You lack confounder analyses like BMI, diabetic status etc which may have affected the surgical outcome and most importantly apparently all these women underwent surgery by different surgical teams. So in my opinion there are simply too many factors that are not controlled for in this very small cohort to draw any conclusions from it and therefore in my opinion this needs to be presented very differently without any conclusions.
Additionally, your flow chart lacks numbers.
Kind regards, Reviewer 3
This manuscript is a resubmission of an earlier submission. The following is a list of the peer review reports and author responses from that submission.
Round 1
Reviewer 1 Report
Please see attached.
